# Associations of Cognitive Function with Serum Magnesium and Phosphate in Hemodialysis Patients: A Cross-Sectional Analysis of the Osaka Dialysis Complication Study (ODCS)

**DOI:** 10.3390/nu16213776

**Published:** 2024-11-03

**Authors:** Tetsuo Shoji, Katsuhito Mori, Yu Nagakura, Daijiro Kabata, Kaori Kuriu, Shinya Nakatani, Hideki Uedono, Yuki Nagata, Hisako Fujii, Yasuo Imanishi, Tomoaki Morioka, Masanori Emoto

**Affiliations:** 1Department of Vascular Medicine, Osaka Metropolitan University Graduate School of Medicine, Osaka 545-8585, Japan; yuki.nagata@omu.ac.jp; 2Vascular Science Center for Translational Research, Osaka Metropolitan University Graduate School of Medicine, Osaka 545-8585, Japan; emoto-m@omu.ac.jp; 3Department of Nephrology, Osaka Metropolitan University Graduate School of Medicine, Osaka 545-8585, Japan; ktmori@omu.ac.jp; 4Department of Metabolism, Endocrinology and Molecular Medicine, Osaka Metropolitan University Graduate School of Medicine, Osaka 545-8585, Japan; c21228s@omu.ac.jp (Y.N.); nakatani-s@omu.ac.jp (S.N.); uedono1217@omu.ac.jp (H.U.); imanishig@gmail.com (Y.I.); moriokatmed@omu.ac.jp (T.M.); 5Center for Mathematical and Data Sciences, Kobe University, Kobe 657-8501, Japan; daijiro.kabata@port.kobe-u.ac.jp; 6Department of Medical Statistics, Osaka Metropolitan University Graduate School of Medicine, Osaka 545-8585, Japan; a19ma037@st.osaka-cu.ac.jp; 7Department of Health and Medical Innovation, Osaka Metropolitan University Graduate School of Medicine, Osaka 545-8585, Japan; hfujii@omu.ac.jp

**Keywords:** cognition, dementia, magnesium, phosphate, mineral bone disorder, hemodialysis

## Abstract

Cognitive impairment and dementia are common in patients with chronic kidney disease, including those undergoing hemodialysis. Since magnesium and phosphate play important roles in brain function and aging, alterations in these and other factors related to bone mineral disorder (MBD) may contribute to low cognitive performance in patients on hemodialysis. This cross-sectional study examined the associations between cognitive function and MBD-related factors among 1207 patients on maintenance hemodialysis. Cognitive function was assessed by the Modified Mini-Mental State examination (3MS). The exposure variables of interest were serum magnesium, phosphate, calcium, calcium–phosphate product, intact parathyroid hormone, fetuin-A, T50 calciprotein crystallization test, use of phosphate binders, use of cinacalcet, and use of vitamin D receptor activators. Multivariable-adjusted linear regression models were used to examine the associations between 3MS and each of the exposure variables independent of 13 potential non-mineral confounders. We found that lower 3MS was associated with lower serum magnesium, lower phosphate, lower calcium–phosphate product, and nonuse of phosphate binders. These results suggest that magnesium and phosphate play potentially protective roles against cognitive impairment in this population.

## 1. Introduction

Low cognitive performance and dementia are prevalent in individuals with chronic kidney disease (CKD) [1,2,3,4], particularly in those undergoing hemodialysis [1,5,6]. Kurella et al. showed that patients with end-stage kidney disease on hemodialysis had worse cognitive function as assessed by the Modified Mini-Mental State Examination (3MS) than dialysis-independent patients with CKD [1]. They also reported that dialysis-independent patients with lower kidney function had lower 3MS scores at baseline and greater declines in 3MS scores over four years of follow-up [2]. Various factors are hypothesized to affect the cognition of patients with CKD by vascular and/or neurodegenerative mechanisms [7], and some of the contributing factors are presumably specific to CKD and/or dialysis treatment. Previous studies reported that a lower cognitive performance in CKD patients was associated with a higher age [8], lower education [8], higher systolic blood pressure [9], lower diastolic blood pressure [8,10,11], history of stroke [8], and history of cardiovascular disease other than stroke [12]. Hemodialysis treatment itself could have acute adverse influences on cognition via transient declines in cerebral blood flow [13]. Intradialytic hypotension, increased blood viscosity, and other mechanisms are possibly involved in this process. On the other hand, long-term hemodialysis was associated with lower odds of having new-onset dementia [14], which may be explained by the accelerated removal of amyloid ß by repeated hemodialysis procedures [15].

Bone mineral disorder (MBD) and vascular calcification may play important roles as CKD-related factors influencing cognition. Accelerated aging of klotho-deficient mice is caused by excess phosphate [16,17]. The use of phosphate binders was associated with a lower risk of incident dementia [18], and a higher serum fibroblast growth factor-23 (FGF23) level was cross-sectionally associated with lower cognitive function in patients with kidney failure [19]. Magnesium is another player in MBD, with various functions in health and disease [20,21] including in neurological disorders such as Alzheimer’s disease [22]. Cognitive function is known to be positively associated with magnesium intake in cross-sectional and cohort studies in the general population [23,24,25,26,27]. Animal studies showed that magnesium was protective against cognitive dysfunction, possibly through suppressing hyperexcitability via binding to N-methyl-D-aspartic acid (NMDA) receptors [28] and other receptors [29]. In a mouse model of Alzheimer’s disease, oral administration of magnesium resulted in long-term potentiation of synaptic transmission in the hippocampus [30]. Regarding magnesium in patients undergoing hemodialysis, serum magnesium levels are affected by nutritional intake, magnesium-containing medications such as magnesium oxide [31], intestinal absorption, distribution between intracellular and extracellular compartments, and exchange between blood and dialysate [32], but renal wasting is negligible in patients with negligible urination.

So far, however, information is limited regarding the relationship between cognitive function in CKD and clinical parameters for MBD such as serum levels of magnesium [33,34], phosphate, calcium, and use of medications for MBD [18]. The aim of this study was to explore the possible associations of cognition with these MBD-related parameters in patients on maintenance hemodialysis.

## 2. Materials and Methods

### 2.1. Study Design and Participants

This was a cross-sectional study in patients on maintenance hemodialysis. The outcome was cognitive performance measured with the 3MS. The exposures of interest were serum magnesium, phosphate, calcium, calcium–phosphate product, intact parathyroid hormone (PTH), fetuin-A, T50, use of any phosphate binders, use of any vitamin D receptor activators (VDRAs), and use of cinacalcet. Cinacalcet was the only calcimimetic agent used in Japan at baseline.

The participants in this analysis were selected from 1696 patients on maintenance hemodialysis in Osaka, Japan, who participated in the Osaka Dialysis Complication Study (ODCS). They were enrolled in the year 2012 from seventeen dialysis facilities affiliated with Osaka City University (current Osaka Metropolitan University) in Osaka Prefecture as described elsewhere [8]. The inclusion and exclusion criteria for the ODCS were described previously [8]. From all the participants of the ODCS, we selected eligible participants for this analysis by excluding those who were missing the outcome variable (3MS) and/or the ten key exposure variables as indicated above.

The ODCS adhered to the Declaration of Helsinki, and the study protocol was reviewed and approved by the Ethics Committee, Osaka City University Graduate School of Medicine, Osaka, Japan (Approval No. 2219), and registered at UMIN-CTR (UMIN000007470). All the participants gave written informed consent before participation in this study.

### 2.2. Measurement of Cognitive Performance Using 3MS

We used a Japanese-translated version of the 3MS, which was originally developed in English by Teng et al. [35], as reported elsewhere [8]. The 3MS evaluates global cognitive function including eight cognitive domains, namely “registration and recall”, “long-term memory”, “orientation”, “attention”, “verbal fluency and understanding”, “word retrieval”, “visuospatial skills”, and “similarities”. We conducted cognitive function tests within the first hour of a hemodialysis session to avoid the possible influence of hypotension during dialysis. Patients were examined sitting or lying on the dialysis bed with partitions between beds if needed. An attending nurse asked questions and recorded the responses of the patient. The nurses at the 17 dialysis facilities took a 1 h lecture by a registered clinical psychologist (Ms. Ayumi Yokote, Department of Neurology, Osaka City University Hospital, Osaka, Japan) to standardize the examination procedure.

We used the 3MS in this study because the 3MS was used in important previous studies in patients with CKD and those on dialysis [1,2,36]. Although it is not the best for the screening of severe cognitive impairment in hemodialysis patients [37], we considered the 3MS one of the most useful tests for the early detection of cognitive impairment.

### 2.3. Measurements of MBD-Related Factors

Blood samples were collected immediately before starting the hemodialysis session on the first day of the week (Monday or Tuesday). Serum magnesium, phosphate, calcium, and intact PTH levels were measured by standard clinical laboratory methods. Calcium–phosphate product was calculated. Fetuin-A was measured using freshly frozen serum samples and stored at −80 °C by enzyme-linked immunoassays using a commercially available kit (BioVender Laboratory Medicine, Modrice, Czech Republic) as previously described [38,39,40]. The T50 calciprotein crystallization test (T50 in short) was conducted at our laboratory according to the method by Pasch et al. [41] by using serum samples which were freshly frozen and stored at −80 °C. The procedure of the T50 assay at our laboratory was previously described elsewhere in detail [42,43]. We confirmed the stability of the T50 assay by showing that the intra-assay and inter-assay coefficients of variation were less than 4.5% [43], and it was validated against the gold standard measurement at Calciscon AG (Nidau, Switzerland) [43]. We used the average of the duplicated measurements of T50 for analysis.

### 2.4. Covariates

Based on previous studies, we considered the following 13 variables as potential confounders: age, sex, dialysis vintage, underlying kidney disease (diabetic kidney disease or not), prior stroke, prior non-stroke cardiovascular disease, education level (lower than college versus college or higher), systolic blood pressure, diastolic blood pressure, blood hemoglobin, body mass index, serum albumin, and C-reactive protein (log-transformed). In the ODCS, cardiovascular disease was defined as coronary artery disease (myocardial infarction and/or coronary revascularization), stroke (ischemic and/or hemorrhagic, with a sudden onset of neurological symptoms, confirmed by a brain imaging test such as computed tomography and magnetic resonance, and excluding transient ischemic attack), peripheral artery disease (amputation, percutaneous angioplasty, and/or bypass grafting as treatment for an ischemic limb, excluding leg pain without such treatment), and congestive heart failure (pulmonary edema requiring hospitalization, excluding dyspnea due to pneumonia and other respiratory diseases) as described elsewhere [8].

### 2.5. Statistical Analysis

The distribution of the outcome variable 3MS score was shown using a histogram and summarized by the median and 25th and 75th percentile levels [interquartile range, IQR]. We summarized the baseline characteristics of participants of this study, which were divided into four groups according to quartiles of 3MS scores, using median [IQR] for continuous variables and number (percentage) for categorical variables for the four groups. Differences across quartiles were evaluated by the Kruskal–Wallis test and χ^2^ test as appropriate.

The association of 3MS scores with each of the ten exposures of interest was examined by unadjusted and multivariable-adjusted linear regression analysis. Adjustment was carried out for the above-mentioned 13 covariates. To meet normal residual assumption, 3MS scores were mathematically transformed by the following equation before entering the model:3MS’ = 2 − Log_10_(101 − 3MS)

By this transformation, a 3MS score of 100 (full mark) is transformed into 2, a 3MS score of 1 is transformed into 0, and the median 3MS score of 91 is transformed into 1.

As a sensitivity analysis, we assessed the nonlinear association between the 3MS and each of the seven continuous variables of interest using a restricted-cubic-spline function with three knots in the multivariable regression model.

Because we found that some of the exposure variables of interest were associated with cognitive function, we conducted four sets of additional analyses. First, to examine whether these associations were simply reflecting poor cognitive performance due to malnutrition, body mass index and serum albumin in the above-mentioned 13 covariates were replaced by the Geriatric Nutritional Risk Index (GNRI) [44,45] or by the Nutritional Risk Index for Japanese Hemodialysis Patients (NRI-JH) [46]. Also, we examined a possible effect modification by nutritional status by inserting the interaction term between the exposure variable and each of these nutritional indexes. If the effect modification was found to be significant, we conducted a subgroup analysis. Second, to examine whether age was an effect modifier of the observed associations, we conducted analysis by inserting the interaction term between age and the factor. If the interaction term was significant, stratified analysis by age (<65 versus ≥65 years) was conducted. Third, to examine the possible influence of including patients with residual kidney function and urination, we tested an effect modification by dialysis vintage on the associations between cognitive function and exposure in the total population. Also, we examined the associations of cognitive function with the exposure variable, excluding those with dialysis vintage < 2 years. And fourth, to address the possible influence of dialysis modality, patients treated with hemodiafiltration were excluded from the analysis.

These statistical calculations were conducted using statistical software JMP 14.3.0 (SAS Institute Japan, Tokyo, Japan) and R version 4.3.2 (The R Foundation for Statistical Computing, Vienna, Austria). A *p*-value < 0.05 by a two-sided test was considered statistically significant. Since the proportion of missing data was small, we analyzed complete data without imputation.

## 3. Results

### 3.1. Selection of Participants

Figure 1 shows the selection of participants for this analysis. Of the 1696 participants of the ODCS, 1207 patients were selected for this study by excluding 469 and 20 patients because of missing 3MS values and the key exposure variables, respectively.

### 3.2. Characteristics of the Participants

Figure 2 shows the distribution of 3MS scores in all the participants in this analysis. It was skewed with a median of 91. Table 1 summarizes the characteristics of the patients by quartiles of 3MS scores. Patients with a lower 3MS score had lower levels of serum phosphate, calcium, calcium–phosphate product, magnesium, fetuin-A, and T50, and lower proportions of the use of phosphate binders and cinacalcet. Also, patients with a lower 3MS score had older age, a shorter duration of dialysis, a higher proportion of diabetic kidney disease, a higher proportion of prior stroke, a lower education level, a lower diastolic blood pressure, and a lower serum albumin level.

### 3.3. Unadjusted Associations of 3MS with MBD-Related Parameters

Table 2 summarizes unadjusted associations of 3MS with the ten MBD-related parameters of interest by linear regeneration analysis. The 3MS showed positive associations with serum magnesium, phosphate, calcium, calcium–phosphate product, fetuin-A, and T50. The 3MS showed positive associations with the use of phosphate binders and the use of cinacalcet.

### 3.4. Adjusted Associations of 3MS with MBD-Related Parameters

Table 3 shows that serum magnesium was positively associated with 3MS scores independent of the covariates. Among the 13 covariates, age, prior stroke, prior non-stroke cardiovascular disease, and systolic blood pressure were inversely associated with 3MS scores, whereas dialysis vintage, higher levels of education, and diastolic blood pressure were positively associated with 3MS scores.

Table 4 shows that serum phosphate was positively associated with 3MS scores by multivariable-adjusted linear regression analysis. Table 5 summarizes the key results obtained from the ten separate multivariable-adjusted linear regression analyses of the factors of interest in association with 3MS scores. When adjusted, 3MS scores did not show a significant association with serum calcium, intact PTH, fetuin-A, T50, the use of cinacalcet, or the use of VDRAs, although the association between 3MS and the use of phosphate binders was at a borderline significance (*p* = 0.051). Figure 3 is a graphical presentation of the key results of the seven continuous exposures of interest with 3MS scores by multivariable-adjusted linear regression.

Appendix A gives the results of a sensitivity analysis showing the associations between the seven measurements of interest and 3MS scores in which a nonlinear association was considered. The positive association between serum magnesium and 3MS score was steeper in the lower range of serum magnesium, whereas the association was less evident in the higher range of serum magnesium, and the overall association was less significant (*p* = 0.052).

The first set of additional analyses examined whether the observed associations of 3MS score with serum magnesium, phosphate, and calcium–phosphate product were attributable to nutritional status by replacing body mass index and serum albumin included as covariates in the original statistical model with either GNRI or NRI-JH. We found that the positive associations of the 3MS score with serum magnesium, phosphate, and calcium–phosphate product remained significant and independent of these nutritional indexes (*p*-values < 0.045 for all). NRI-JH was not a significant effect modifier on the relationship of 3MS score with serum magnesium, phosphate, or calcium–phosphate product (*p* for interaction = 0.584, 0.127, and 0.183, respectively). Also, GNRI was not a significant effect modifier on the relationship of the 3MS score with serum magnesium, whereas GNRI significantly modified the associations of the 3MS score with serum phosphate and calcium–phosphate product (*p* for interaction = 0.030 and 0.036, respectively). Subgroup analysis revealed that the positive associations of 3MS score with serum phosphate and calcium–phosphate product were significant in the subgroup with GNRI levels lower than the median (*p* = 0.011 and *p* = 0.012, respectively), whereas these associations were not significant in the subgroup with higher GNRI levels (*p* = 0.844 and *p* = 0.475, respectively).

The second set of additional analyses examined whether the observed associations of the 3MS score with serum magnesium, phosphate, and calcium–phosphate product were modified by age. We found that the interaction term between age and serum magnesium was not significant (*p* for interaction = 0.251), whereas the interaction terms between age and serum phosphate (*p* for interaction = 0.003) and between age and calcium–phosphate product (*p* for interaction = 0.019) were significant. Stratified analyses by age revealed that the independent associations of 3MS score with serum phosphate (*p* = 0.010) and calcium–phosphate product (*p* = 0.006) were significant in the elderly group (≥65 years, N = 706), whereas these associations were not significant (*p* = 0.841 and 0.823, respectively) in the non-elderly group (<65 years, N = 501).

The third set of additional analyses tried to address the possible influence of the inclusion of patients with residual kidney function and urination. There was no significant effect modification by dialysis vintage on the association of the 3MS score with serum magnesium (*p* for interaction = 0.126) or phosphate (p for interaction = 0.849). Also, the independent and positive associations of the 3MS score with serum magnesium (*p* = 0.022) and serum phosphate (*p* = 0.032) remained significant when patients with dialysis vintage <2 years were excluded (N = 978).

And the fourth set of additional analyses addressed the possible influence of hemodiafiltration. When 16 patients treated with hemodiafiltration were excluded from the analysis, the independent and positive associations of 3MS scores with serum magnesium (*p* = 0.034) and serum phosphate (*p* = 0.034) remained significant.

## 4. Discussion

This study examined the possible association of cognitive function with each of the ten factors related to MBD in prevalent patients on maintenance hemodialysis. After adjustment, 3MS scores showed positive associations with serum magnesium, serum phosphate, calcium–phosphate product, and the use of phosphate binders, but not significantly with the other six variables of interest in this cross-sectional study.

This study showed a positive association between serum magnesium and 3MS score. This is in line with previous cross-sectional studies in the general population showing a positive association between magnesium intake and cognitive function [23,24]. Also, cohort studies in the general population showed that the risk of dementia was inversely associated with magnesium intake [25], serum magnesium level [26], and the use of magnesium oxide [27]. Although there are no studies investigating the association between magnesium and longitudinal change in cognitive performance in patients with CKD, a recent cross-sectional study in prevalent hemodialysis patients by Yang et al. [33] reported that serum magnesium level showed a U-shaped relationship with the odds of having mild cognitive impairment and that they found the lowest odds in the range of 1.12–1.24 mmol/L (2.72–3.01 mg/dL) of serum magnesium. Another cross-sectional study by Kato et al. [34] reported a significant positive association between serum magnesium levels and cognitive function in 390 elderly patients undergoing hemodialysis using the Montreal Cognitive Assessment and the Mini-Mental State Examination.

It should be noted that 1087 of the 1207 patients (90.1%) of our study had a serum magnesium level of 3.01 mg/dL or lower, indicating that our population was in the left half of the U-shape reported by Yang et al. [33]. In addition, when a nonlinear association was considered between serum magnesium and 3MS score in a sensitivity analysis, the positive association was more evident in the lower range of serum magnesium. Kato et al. [34] reported a positive association between serum magnesium and cognitive performance in elderly (≥65 years) hemodialysis patients. Our additional analysis showed that the positive association between serum magnesium and cognition was not significantly modified by age. Thus, our results not only agree well with the reports by Yang et al. [33] and Kato et al. [34] in that lower serum magnesium was an independent factor associated with low cognition in patients undergoing hemodialysis but also extend the association to the non-elderly patient group.

The association between low serum magnesium and low cognition can be explained by the neuroprotective functions of magnesium in addition to its important roles as a cofactor for reactions of adenosine triphosphatases (ATPases) [21]. Magnesium is known to have various functions in cells and systems in health and disease [20] including inflammation and neurological disorders including Alzheimer’s disease [22]. Neuroinflammation is implicated in Alzheimer’s disease in addition to accumulations of amyloid ß and tau tangles, and a low-magnesium status could enhance inflammation and aging (“inflammaging”) [20]. Magnesium can pass the blood–brain barrier, and it is present in the cerebrospinal fluid [47]. Magnesium can be protective against cognitive dysfunction in animal models [28], possibly through suppressing hyperexcitability via binding to N-methyl-D-aspartic acid (NMDA) receptors and other receptors [29]. In a mouse model of Alzheimer’s disease, oral administration of magnesium resulted in the long-term potentiation of synaptic transmission in the hippocampus [30], although the results of animal experiments are not always consistent [48,49]. These experimental data, as well as the above-mentioned observations in the general populations [23,24,25,26,27], support the interpretation that lower serum magnesium levels could have a causative role in lower cognitive performance in patients undergoing hemodialysis.

Regarding the relationship between phosphate and cognitive function, this study showed that the 3MS score was positively associated with serum phosphate, calcium–phosphate product, and the use of phosphate binders. Because no significant association was found between the 3MS score and serum calcium, the positive association of the 3MS score with calcium–phosphate product is attributable to that with serum phosphate. The latter result regarding phosphate binder use is consistent with a recent report by Mathur et al. [18], in which the use of treatment for MBD was associated with a lower risk of incident dementia, although they did not report the association of laboratory data with the incidence of dementia. Studies by us and by Mathur et al. [18] suggest that cognitive function was higher or maintained in those who needed phosphate binders as a treatment for hyperphosphatemia. On the contrary, our results appear to be inconsistent with the report by Drew et al. showing an inverse association between FGF23 and cognitive performance [19], when based on the known positive correlation between FGF23 and serum phosphate. However, since FGF23 was reported to inhibit intestinal magnesium absorption in rats [50], the inverse association between FGF23 and cognition may be mediated by magnesium.

It is an important question whether these associations of cognitive performance with hypophosphatemia and hypomagnesemia are simply attributable to general malnutrition. The original statistical model was adjusted for the two nutritional variables, namely body mass index and serum albumin, in addition to the other 11 covariates, and we observed the significant associations of cognitive function with serum phosphate and serum magnesium which were independent of these nutritional variables. The results of an additional analysis in which body mass index and serum albumin were replaced with GNRI or NRI-JH again showed that the positive associations of cognitive function with serum phosphate and magnesium were significant and independent of these nutritional risk indexes. Thus, we interpret these results to indicate the importance of serum levels of phosphate and magnesium rather than general nutritional status in cognitive impairment among patients undergoing hemodialysis.

This study revealed that age and GNRI were significant effect modifiers on the positive association between serum phosphate and 3MS score. The association was significant in the age group of ≥65 years and the subgroup with lower GNRI levels, whereas it was not significant in the age group of <65 years or in the subgroup with higher GNRI levels. These findings suggest that a low serum phosphate level is particularly important in cognitive impairment among patients of higher ages and those with poorer nutrition.

How do we interpret the observed positive association between serum phosphate and cognitive performance? It appears to be against our hypothesis that the toxicity of phosphate excess could be causatively involved in low cognitive function in hemodialysis patients. However, this result can be alternatively interpreted to indicate that lower serum phosphate was associated with lower cognitive performance. Serum phosphate is lower in hemodialysis patients of older age [51]. Patients on hemodialysis with a lower serum phosphate level are known to be at a higher risk of mortality [51] and cerebral infarction [52]. The real-world observation that the nonuse of medications for MBD including phosphate binders was predictive of a higher risk of incident dementia [18] can be interpreted to indicate that those who needed no phosphate binders, possibly because of low serum phosphate, were at a higher risk for dementia. Therefore, these studies support the notion that a low-phosphate condition could have a negative influence on cognition in patients with kidney failure. Phosphate is an integral component of ATP, which is an essential energy source for all cells including the central nervous system. This situation of phosphate may be similar to the relationship between hypoglycemia and brain functions. Hypophosphatemia was shown to reduce the rates of muscle ATP synthesis in mice [53]. Thus, we speculate that the observed associations of cognitive function with serum phosphate levels could indicate that hypophosphatemia is unfavorable for cognitive performance in this population, presumably due to impaired energy metabolism. Based on our additional analysis, elderly patients might be more susceptible to hypophosphatemia than their non-elderly counterparts.

It is important to note that serum phosphate is usually measured immediately before the hemodialysis session in routine clinical practice. Since dialysate contained no phosphate, as shown in Appendix A, patients with lower predialysis serum phosphate levels (for example, between 3.0 and 3.5 mg/dL) are more likely to experience severe postdialysis hypophosphatemia (lower than 2.0 mg/dL) [54], and this could be repeated by regular hemodialysis. In this study, the proportion of patients with predialysis serum phosphate levels < 3.5 mg/dL was highest in the lowest quartile of 3MS scores as shown in Appendix A. A recent study revealed that the QT interval on electrocardiograms shows a U-shaped relationship with serum predialysis phosphate concentration in patients undergoing hemodialysis, indicating that hypophosphatemia is associated with QT interval prolongation [54], which can be explained by poor ATP-derived energy generation in the myocardium. Although there is no direct evidence for the influence of hypophosphatemia on brain functions, it is possible that repeated severe postdialysis hypophosphatemia could have an adverse influence on cognitive function in patients with low predialysis serum phosphate concentrations.

We found no significant associations of 3MS scores with the other MBD-related factors such as serum calcium, intact PTH, fetuin-A, T50, the use of cinacalcet, and the use of VDRAs. Among these, T50 is known to associate positively with serum magnesium and inversely with serum phosphate. Therefore, no significant association between T50 and 3MS score could be explained by the counteracting effects of magnesium and phosphate on T50. The result regarding the use of cinacalcet and the use of VDRAs in this study was consistent with the result of a randomized controlled trial with a calcimimetic agent etelcalcetide and a VDRA maxacalcitol for 12 months, which showed no change in cognition in either group of hemodialysis patients with secondary hyperparathyroidism [43]. A higher plasma fetuin-A concentration was associated with better cognitive performance in patients with mild-to-moderate Alzheimer’s disease [55], and higher fetuin-A was associated with a lower risk of cognitive decline during follow-up for 4 years in community-dwelling older adults [56]. Our results on fetuin-A were inconsistent with these previous studies, presumably because of the differences in the study design and populations.

This study has several limitations. First, this study included only hemodialysis patients in Japan; therefore, the generalizability should be confirmed in other populations. Second, the results were based on single measurements which could overestimate or underestimate the true associations. Third, although we reported the associations of cognitive function with serum magnesium and phosphate, given the wide variety of foods and medications with magnesium oxide, it was not possible to obtain data regarding consumed dietary and drinking water intake of magnesium, or supplements containing magnesium. Although dietary guidance was given according to the guidelines of the Japanese Society of Nephrology 2007, which recommended a protein intake between 1.0 and 1.2 g/kg body weight per day and a dietary phosphate intake of 15 mg per 1 g of protein per day for patients with CKD stage 5D treated with dialysis, we did not measure the dietary intake of phosphate. Fourth, we used serum magnesium, phosphate, and other serum parameters immediately before starting the hemodialysis session. This may have affected the results, because some factors such as magnesium and phosphate fluctuated following meals and hemodialysis sessions. The lack of postdialysis values and trajectories of these factors between dialysis sessions is another limitation, and such information may provide further insight into the mechanisms of cognitive impairment in patients undergoing hemodialysis. Fifth, because of the cross-sectional nature of this study, the observed associations do not necessarily indicate causality. Sixth, this study did not consider all the medications, electrolytes, and cardiac function as evaluated by echocardiography, biomarkers such as N-terminal pro-B-type natriuretic peptide, and other comorbidities that could affect the nervous system and cognitive function. Seventh, this study used serum total magnesium concentration, and the lack of ionized magnesium may be another limitation. Eighth, because some medications could result in simultaneous changes in related variables, caution should be exercised in the interpretation of the results of associations. Ninth, this study included a small number of patients (16 out of 1207) who were treated with hemodiafiltration in which infusion and ultrafiltration were conducted at a rate of 12 L per hour using infusate with the same composition as the dialysate. Because this fact might have affected the results, we confirmed the results by conducting an additional analysis in which patients treated with hemodiafiltration were excluded. On the other hand, this study included a relatively large number of hemodialysis patients, which allowed the statistical adjustment of various covariates. This is one of the strengths of this study. In addition, we revealed that age and GNRI modified the association between serum phosphate and cognition for the first time in patients undergoing hemodialysis. These findings are novel and another strength of this study.

## 5. Conclusions

In conclusion, this cross-sectional study of hemodialysis patients revealed that lower levels of serum magnesium and serum phosphate were associated with lower cognitive function, suggesting a potential link between these MBD-related factors and cognition. Age and GNRI were significant effect modifiers on the relationship between cognition and phosphate but not on the relationship between cognition and magnesium. These results suggest that phosphate and magnesium play potentially different roles in cognitive impairment, and that the role of phosphate in cognitive function may vary according to patient age and nutritional status. Clearly, further longitudinal studies and trials are needed to establish the causality and clinical relevance of the findings of this study.

## Figures and Tables

**Figure 1 nutrients-16-03776-f001:**
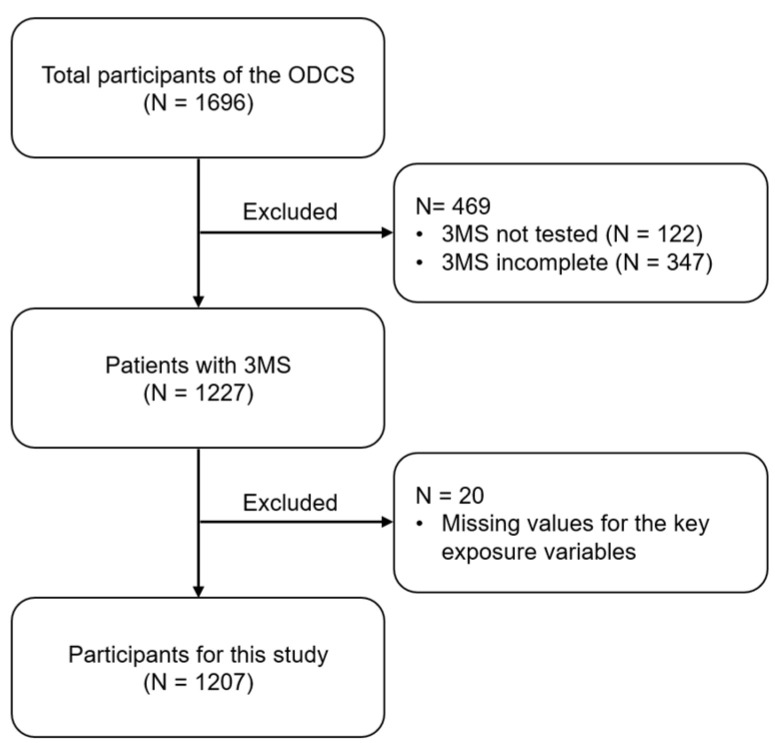
Selection of the study participants. We excluded patients with missing 3MS values and the nine key exposure variables, and the remaining 1207 patients were selected for this study. Abbreviations: ODCS, Osaka Dialysis Complication Study; 3MS, Modified Mini-Mental State examination.

**Figure 2 nutrients-16-03776-f002:**
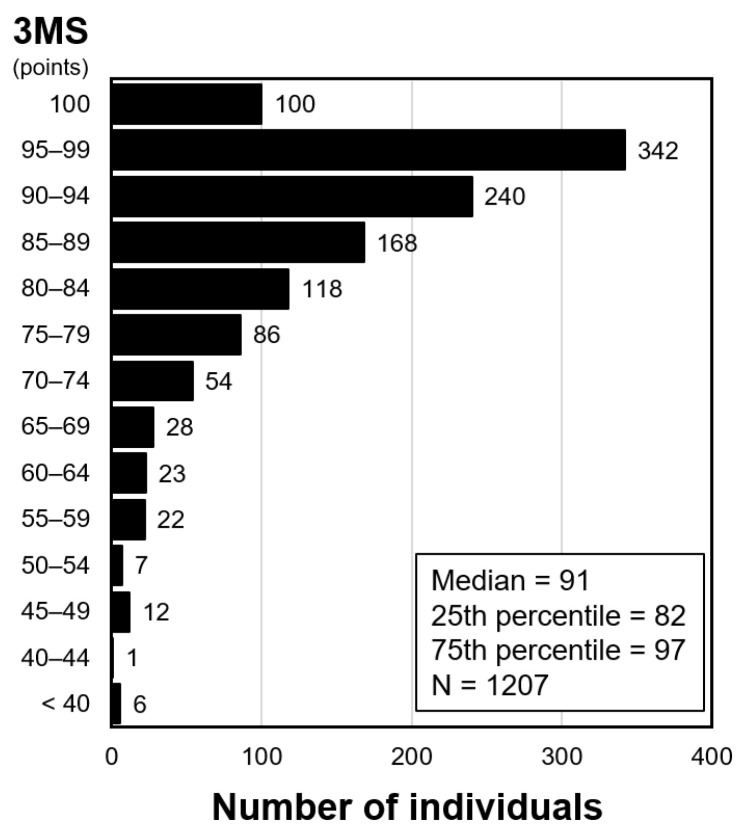
Distribution of 3MS scores. Abbreviations: 3MS, Modified Mini-Mental State examination.

**Figure 3 nutrients-16-03776-f003:**
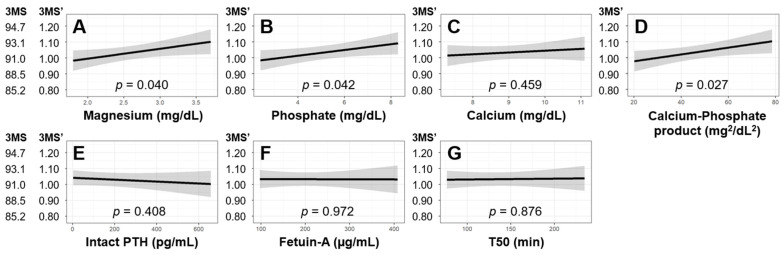
Associations of 3MS scores with seven MBD-related serum parameters by multivariable-adjusted linear regression models. Associations of 3MS scores and MBD-related factors were examined by multivariable-adjusted linear regression analysis. The 3MS score was mathematically transformed before entering the model: 3MS’ = 2 − Log10(101 − 3MS). Panels (**A**)–(**G**) show the results for serum magnesium, phosphate, calcium, calcium–phosphate product, intact PTH, fetuin-A, and T50, respectively. Solid lines and shaded areas are regression lines and 95% confidence intervals. Abbreviations: 3MS, Modified Mini-Mental State examination; PTH, parathyroid hormone; T50, T50 calciprotein crystallization test.

**Table 1 nutrients-16-03776-t001:** Characteristics of participants by 3MS quartiles.

Variable	Unit	Total (N = 1207)	Q1 (N = 309)	Q2 (N = 299)	Q3 (N = 356)	Q4 (N = 243)	*p*-Value	Missing, N (%)
3MS	point	91 [82, 97]	75 [65, 79]	88 [86, 90]	95 [93, 96]	99 [98, 100]	<0.001	0 (0%)
Mg	mg/dL	2.6 [2.4, 2.8]	2.5 [2.3, 2.7]	2.6 [2.3, 2.8]	2.6 [2.4, 2.9]	2.7 [2.4, 2.9]	<0.001	0 (0%)
Phosphate	mg/dL	5.1 [4.3, 5.9]	4.8 [4, 5.5]	5.1 [4.4, 6]	5.2 [4.5, 6]	5.4 [4.5, 6.3]	<0.001	0 (0%)
Calcium	mg/dL	9.0 [8.5, 9.5]	8.9 [8.4, 9.3]	8.9 [8.4, 9.4]	9.1 [8.6, 9.5]	9.1 [8.6, 9.6]	<0.001	0 (0%)
Ca-P product	mg^2^/dL^2^	45.9 [38.3, 53.9]	42.6 [34.4, 50.3]	45.9 [38.7, 54.3]	47.3 [40.1, 55.1]	49.5 [41.0, 56.4]	<0.001	0 (0%)
intact PTH	pg/mL	116 [60, 190]	114 [61, 169]	110 [61, 194]	114 [53, 200]	137 [70, 209]	0.108	0 (0%)
Fetuin-A	µg/mL	186 [156, 221]	175 [150, 211]	182 [154, 219]	188 [162, 222]	194 [164, 226]	<0.001	0 (0%)
T50	min	128 [109, 151]	122 [103, 146]	126 [108, 148]	130 [110, 153]	134 [113, 159]	<0.001	0 (0%)
Use of P-binders	N (%)	1062 (88.0%)	247 (79.9%)	254 (85.0%)	331 (93.0%)	230 (94.7%)	<0.001	0 (0%)
Use of VDRAs	N (%)	864 (71.6%)	207 (67.0%)	207 (69.2%)	273 (76.7%)	177 (72.8%)	0.033	0 (0%)
Use of cinacalcet	N (%)	249 (20.6%)	45 (14.6%)	47 (15.7%)	87 (24.4%)	70 (28.8%)	<0.001	0 (0%)
Age	year	67 [60, 74]	73 (68, 78)	70 (64, 75)	64 (56, 71)	60 (50, 65)	<0.001	0 (0%)
Male sex	N (%)	766 (63.5%)	197 (63.8%)	193 (64.6%)	218 (61.2%)	158 (65.0%)	0.758	0 (0%)
Dialysis vintage	year	5.8 [2.5, 11.8]	4.5 [2.3, 9.6]	5.0 [2.2, 9.7]	6.6 [2.8, 13.8]	8.8 [3.4, 16.5]	<0.001	0 (0%)
DKD	N (%)	473 (39.2%)	141 (45.6%)	130 (43.5%)	130 (36.5%)	72 (29.6%)	<0.001	0 (0%)
Prior stroke	N (%)	207 (17.1%)	81 (26.2%)	58 (19.4%)	49 (13.8%)	19 (7.8%)	<0.001	0 (0%)
Prior CAD	N (%)	166 (13.8%)	56 (18.1%)	52 (17.4%)	39 (11.0%)	19 (7.8%)	<0.001	0 (0%)
Prior PAD	N (%)	77 (6.4%)	21 (6.8%)	26 (8.7%)	21 (5.9%)	9 (3.7%)	0.120	0 (0%)
Prior CHF	N (%)	150 (12.4%)	54 (17.5%)	42 (14.1%)	39 (11.0%)	15 (6.2%)	<0.001	0 (0%)
Prior non-stroke CVD	N (%)	228 (18.9%)	67 (21.7%)	73 (24.4%)	57 (16.0%)	31 (12.8%)	0.002	0 (0%)
Prior any CVD	N (%)	435 (36.0%)	148 (47.9%)	131 (43.8%)	106 (29.8%)	50 (20.6%)	<0.001	0 (0%)
Education (≥univ)	N (%)	244 (20.2%)	32 (10.4%)	46 (15.4%)	83 (23.3%)	83 (34.2%)	<0.001	0 (0%)
SBP	mmHg	150 [134, 164]	150 [133, 163]	150 [138, 164]	148 [133, 164]	148 [132, 164]	0.442	8 (0.7%)
DBP	mmHg	75 [68, 83]	70 [66, 78]	74 [67, 81]	78 [69, 84]	79 [69, 89]	<0.001	13 (1.1%)
Hemoglobin	g/dL	10.6 [10.0, 11.3]	10.5 [9.8, 11.3]	10.6 [10, 11.3]	10.7 [10, 11.3]	10.7 [10.1, 11.4]	0.110	0 (0%)
Body mass index	kg/m^2^	21.2 [19.2, 23.6]	21.1 [18.9, 23.3]	21.2 [19.2, 23.3]	21.2 [19.4, 23.8]	21.4 [19.3, 23.9]	0.343	0 (0%)
Albumin	g/dL	3.7 [3.5, 3.9]	3.7 [3.4, 3.8]	3.7 [3.5, 3.9]	3.8 [3.6, 4]	3.8 [3.6, 4]	<0.001	0 (0%)
C-reactive protein	mg/dL	0.10 [0.05, 0.29]	0.12 [0.05, 0.40]	0.1 [0.05, 0.29]	0.10 [0.05, 0.25]	0.10 [0.05, 0.19]	0.137	0 (0%)
Sodium	mEq/L	139 [137, 141]	139 [137, 141]	140 [138, 141]	140 [137, 141]	139 [137, 141]	0.005	0 (0%)
Potassium	mEq/L	4.7 [4.2, 5.2]	4.6 [4.1, 5.0]	4.7 [4.2, 5.2]	4.7 [4.3, 5.2]	4.8 [4.4, 5.4]	<0.001	0 (0%)
Chloride	mEq/L	102 [99, 105]	101 [99, 104]	103 [100, 105]	102 [100, 105]	102 [100, 106]	<0.001	0 (0%)
GNRI	point	95.9 [90.8, 101.8]	94.6 [87.7, 100.3]	95.1 [90.7, 100.9]	96.7 [92.0, 102.7]	97.5 [92.1, 103.6]	<0.001	0 (0%)
NRI-JH	point	3 [0, 5]	4 [0, 7]	3 [0, 5]	3 [0, 5]	3 [0, 4]	0.004	0 (0%)
Dialysis frequency								
2 sessions per week	N (%)	28 (2.3%)	11 (3.6%)	5 (1.7%)	8 (2.2%)	4 (1.7%)		
3 sessions per week	N (%)	1173 (97.3%)	295 (95.8%)	294 (98.3%)	346 (97.2%)	238 (98.3%)		
4 sessions per week	N (%)	4 (0.3%)	2 (0.6%)	0 (0.0%)	2 (0.6%)	0 (0.0%)	0.368	2 (0.2%)
Dialysis time per session	hours	4.0 [4.0, 4.0]	4.0 [3.5, 4.0]	4.0 [3.5, 4.0]	4.0 [4.0, 4.0]	4.0 [4.0, 4.0]	<0.001	7 (0.6%)
UFR	mL/kg/h	12.9 [10.0, 15.9]	12.9 [9.8, 16.5]	12.8 [9.6, 15.4]	13.4 [10.3, 16.1]	12.4 [9.9, 15.5]	0.122	13 (1.1%)
spKt/V	no unit	1.38 [1.20, 1.59]	1.37 [1.20, 1.56]	1.37 [1.12, 1.54]	1.37 [1.20, 1.60]	1.40 [1.20, 1.62]	0.048	70 (5.8%)
D-Ca 2.5 mEq/L	N (%)	186 (15.4%)	34 (11.0%)	49 (16.4%)	65 (18.3%)	38 (15.6%)		
D-Ca 2.75 mEq/L	N (%)	304 (25.2%)	41 (13.3%)	63 (21.1%)	94 (26.4%)	106 (43.6%)		
D-Ca 3.0 mEq/L	N (%)	717 (59.4%)	234 (75.3%)	187 (62.5%)	197 (55.3%)	99 (70.7%)	<0.001	0 (0%)
D-Mg 1.0 mEq/L	N (%)	1207 (100%)	309 (100%)	299 (100%)	356 (100%)	243 (100%)	1.000	0 (0%)
Hemodiafiltration	N (%)	16 (1.3%)	4 (1.3%)	5 (1.7%)	5 (1.4%)	2 (0.8%)	0.858	0 (0%)

The table gives medians [25th and 75th percentile levels] for continuous variables and numbers (percentages) for binary variables for quartiles of 3MS. Differences across quartiles were evaluated by the Kruskal–Wallis test and χ^2^ test as appropriate. Abbreviations: 3MS, Modified Mini-Mental State examination; Mg, magnesium; Ca-P product, calcium–phosphate product; PTH, parathyroid hormone; T50, T50 calciprotein crystallization test; P-binder, phosphate binder; VDRA, vitamin D receptor activator; DKD, diabetic kidney disease; CAD, coronary artery disease; PAD, peripheral artery disease; CHF, congestive heart failure; CVD, cardiovascular disease; Education (≥univ), education of university and higher level; SBP, systolic blood pressure; DBP, diastolic blood pressure; GNRI, Geriatric Nutritional Risk Index; NRI-JH, Nutritional Risk Index for Japanese Hemodialysis Patients; UFR, ultrafiltration rate; spKt/V, single-pool Kt/V; D-Ca, dialysate calcium; D-Mg, dialysate magnesium; Q1−Q4, quartile 1−4.

**Table 2 nutrients-16-03776-t002:** Summary of the unadjusted associations of the 3MS with each of the ten MBD-related factors which were separately analyzed.

Variable	Unit	Coefficient	Lower 95%	Higher 95%	*p*-Value
Magnesium	0.4 mg/dL	0.074	0.047	0.101	<0.001
Phosphate	1.6 mg/dL	0.094	0.062	0.125	<0.001
Calcium	1.0 mg/dL	0.072	0.038	0.106	<0.001
Calcium–phosphate product	15.7 mg^2^/dL^2^	0.116	0.083	0.149	<0.001
Intact PTH	130 pg/mL	0.021	−0.000	0.043	0.054
T50	41.9 min	0.060	0.029	0.091	<0.001
Fetuin-A	65 µg/dL	0.042	0.014	0.070	0.003
Use of P-binders	yes = 1, no = 0	0.131	0.093	0.170	<0.001
Use of VDRAs	yes = 1, no = 0	0.028	−0.000	0.056	0.054
Use of cinacalcet	yes = 1, no = 0	0.074	0.042	0.105	<0.001

These data were derived from ten unadjusted linear regression models. The 3MS score was mathematically transformed before entering the model: 3MS’ = 2 − Log10(101 − 3MS). Coefficients (lower and higher 95% confidence intervals) for the continuous exposure variables were calculated for the interquartile range so that comparison among variables could be easier. Abbreviations: 3MS, Modified Mini-Mental State examination; PTH, parathyroid hormone; T50, T50 calciprotein crystallization test; VDRA, vitamin D receptor activator.

**Table 3 nutrients-16-03776-t003:** Independent association of 3MS scores with serum magnesium adjusted for the 13 potential confounders.

Variable	Unit	Coefficient	Lower 95%	Higher 95%	*p*-Value
Age	year	−0.015	−0.017	−0.012	<0.001
Sex	male = 1, female = 0	−0.001	−0.024	0.022	0.916
Dialysis vintage	year	0.009	0.006	0.012	<0.001
Diabetic kidney disease	yes = 1, no = 0	−0.015	−0.039	0.010	0.242
Prior stroke	yes = 1, no = 0	−0.066	−0.095	−0.036	<0.001
Prior non-stroke CVD	yes = 1, no = 0	−0.033	−0.062	−0.005	0.022
Education (≥Univ.)	yes = 1, no = 0	0.097	0.070	0.124	<0.001
Systolic blood pressure	10 mmHg	−0.016	−0.028	−0.003	0.012
Diastolic blood pressure	10 mmHg	0.030	0.008	0.051	0.007
Hemoglobin	g/dL	−0.002	−0.022	0.018	0.876
Albumin	g/dL	0.109	0.034	0.184	0.004
Body mass index	kg/m^2^	0.001	−0.006	0.007	0.831
Log_10_(C-reactive protein)	Log unit	−0.001	−0.042	0.041	0.966
Magnesium	IQR, 0.4 mg/dL	0.025	0.001	0.049	0.040
Coefficient of determination (R^2^) = 0.329 (*p* < 0.001)					

The table gives the result of the multivariable-adjusted linear regression analysis of factors associated with 3MS scores. To meet the normal residual assumption, the 3MS score was mathematically transformed before entering the model: 3MS’ = 2 − Log10(101 − 3MS). Abbreviations: 3MS, Modified Mini-Mental State examination; CVD, cardiovascular disease; IQR, interquartile range.

**Table 4 nutrients-16-03776-t004:** Independent association of 3MS scores with serum phosphate adjusted for the 13 potential confounders.

Variable	Unit	Coefficient	Lower 95%	Higher 95%	*p*-Value
Age	year	−0.014	−0.016	−0.012	<0.001
Sex	male = 1, female = 0	−0.001	−0.024	0.022	0.934
Dialysis vintage	year	0.009	0.006	0.012	<0.001
Diabetic kidney disease	yes = 1, no = 0	−0.012	−0.037	0.012	0.317
Prior stroke	yes = 1, no = 0	−0.066	−0.095	−0.036	<0.001
Prior non-stroke CVD	yes = 1, no = 0	−0.036	−0.064	−0.007	0.014
Education (≥Univ.)	yes = 1, no = 0	0.097	0.070	0.123	<0.001
Systolic blood pressure	10 mmHg	−0.016	−0.028	−0.004	0.009
Diastolic blood pressure	10 mmHg	0.030	0.009	0.052	0.005
Hemoglobin	g/dL	−0.003	−0.023	0.017	0.795
Albumin	g/dL	0.118	0.044	0.192	0.002
Body mass index	kg/m^2^	0.000	−0.007	0.006	0.901
Log_10_(C-reactive protein)	Log unit	−0.007	−0.049	0.035	0.743
Phosphate	IQR, 1.6 mg/dL	0.029	0.001	0.057	0.042
Coefficient of determination (R^2^) = 0.329 (*p* < 0.001)					

The table gives the result of a multivariable-adjusted linear regression analysis of factors associated with 3MS scores. To meet the normal residual assumption, the 3MS score was mathematically transformed before entering the model: 3MS’ = 2 − Log10(101 − 3MS). Abbreviations: 3MS, Modified Mini-Mental State examination; CVD, cardiovascular disease; IQR, interquartile range.

**Table 5 nutrients-16-03776-t005:** Summary of the independent associations of 3MS score with each of the ten MBD-related factors which were separately analyzed.

Variable	Unit	Coefficient	Lower 95%	Higher 95%	*p*-Value	R^2^
Magnesium	0.4 mg/dL	0.025	0.001	0.049	0.040	0.329
Phosphate	1.6 mg/dL	0.029	0.001	0.057	0.042	0.329
Calcium	1.0 mg/dL	0.011	−0.019	0.041	0.459	0.327
Calcium–phosphate product	15.7 mg^2^/dL^2^	0.034	0.004	0.063	0.027	0.329
Intact PTH	130 pg/mL	−0.008	−0.026	0.011	0.408	0.327
Fetuin-A	65 µg/dL	0.000	−0.025	0.024	0.972	0.326
T50	41.9 min	0.002	−0.025	0.030	0.876	0.326
Use of phosphate binders	yes = 1, no = 0	0.035	0.000	0.069	0.051	0.329
Use of VDRAs	yes = 1, no = 0	0.018	−0.006	0.042	0.140	0.328
Use of cinacalcet	yes = 1, no = 0	−0.006	−0.034	0.022	0.662	0.326

These data were derived from ten multivariable-adjusted linear regression models. The 3MS score was mathematically transformed before entering the model: 3MS’ = 2 − Log10(101 − 3MS). Coefficients (lower and higher 95% confidence intervals) for the continuous exposure variables were calculated for the interquartile range so that comparison among variables could be easier. Abbreviations: 3MS, Modified Mini-Mental State examination; PTH, parathyroid hormone; T50, T50 calciprotein crystallization test; VDRA, vitamin D receptor activator; R^2^, coefficient of determination.

## Data Availability

The dataset that supports the findings of this study cannot be shared publicly due to ethical restrictions for the protection of the personal and sensitive information of individuals who participated in the study. The data will be shared on reasonable request to the corresponding author after permission is given by the ethics committee at our institution.

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
