# Peer review of "Associations of Cognitive Function with Serum Magnesium and Phosphate in Hemodialysis Patients: A Cross-Sectional Analysis of the Osaka Dialysis Complication Study (ODCS)"

_nutrients, 2024, doi:10.3390/nu16213776_

Round 1
Reviewer 1 Report (Previous Reviewer 1)
Comments and Suggestions for Authors
The manucript may be published on its present form
Author Response
Pease see the attachment.

Reviewer 2 Report (Previous Reviewer 2)
Comments and Suggestions for Authors
General Comments:
The manuscript provides a well-structured analysis of the association between cognitive function and phosphate, magnesium levels in patients undergoing hemodialysis. The research question is relevant and timely. However, certain points should be addressed.
Major Comments:
1. In chronic hemodialysis patients, the cognitive impairment can be attributed to vascular calcification, as well as ESKD- treatment associated CV factors such as intradialytic hypotension, hyperviscosity and thrombotic events. The authors should provide detailed information to the CAD, PAD, HF history, pro-BNP level, or even latest ejection fraction data over echocardiography other than binary information for stroke history to evaluate the risk factors of cognitive impairment more comprehensively.
2. Although the association between low serum magnesium and low cognition can be explained by neuroprotective functions of magnesium in addition to its important roles as a cofactor for reactions of ATPases. Considering the complexity of HD patients, cross sectional study can only explain the association other than causation and should not be overstated. Low serum Mg and phosphate could be closely related to reduce GI absorption. The authors should further elucidate the relationship between low magnesium and phosphate levels, nutritional status, and cognitive function to determine whether low magnesium and phosphate levels influence cognitive function independently of other nutritional parameters. Subgroup analysis of Mg and phosphate to cognitive function between different nutritional status should be further addressed.
3. The interpretation of results based on calcium, phosphate, and I-PTH levels should be approached with caution, as these parameters may not be appropriate for aggregation across individuals due to their complex interrelationships within each patient, compounded by the potential influence of offending drug use.
Minor Comments:
1. The CKD-MBD associated drug use should be further classified to remote, recent or current drug use since this medication can be adjusted every month in routine clinical visits.

There are several grammatical mistakes that need to be corrected by native speaker to maintain the manuscript's professionalism.
Author Response
Pease see the attachment.

This manuscript is a resubmission of an earlier submission. The following is a list of the peer review reports and author responses from that submission.
Round 1
Reviewer 1 Report
Comments and Suggestions for Authors
Thank you for the opportunity to review this manuscript.
Since preserving cognitive function in patients undergoing maintenance hemodialysis is becoming a relevant, everyday practice clinical issue, the present manuscript represent an important contribution. Some minor issues, however, should be addressed to improve the technical quality of the research.
In the introduction section, briefly mention clinical variables that could influence Magnesium levels in the studied population (i.e. intake, intestinal absorption and renal wasting). Also, mention any other clinical trial using 3MS to assess cognitive function in hemodialysis patients, in order to highlight the external validity of your results (i.e. “3MS has previously been widely used to asses neurocognitive function in HD patients”).
In the Materials and Methods section, specify the chronological relationship between blood sample acquisition and hemodialysis (i.e. immediately before the hemodialysis session). Further into the manuscript, on the discussion section, briefly mention if you consider that blood sample timing (i.e. before/after the hemodyalisis session) may have any influence on your results.
Reviewer 2 Report
Comments and Suggestions for Authors
General Comments:
The manuscript provides a well-structured analysis of the association between cognitive function and phosphate, magnesium levels in patients undergoing hemodialysis. The research question is relevant and timely. However, the manuscript could benefit from a clearer delineation of its findings' clinical implications and potential for influencing current medical practices.
Major Comments:
1. In chronic hemodialysis patients, the cognitive impairment can be more attributed to vascular calcification, as well as ESKD- treatment associated factors such as intradialytic hypotension, hyperviscosity and thrombotic events, intradialytic cerebral ischemia, aside from simply electrolytes imbalance. The authors should provide detailed information to the calcium-phosphorus product, serum Na, K, Cl, pro-BNP level, and latest ejection fraction data over echocardiography to evaluate the risk factors of cognitive impairment more comprehensively.
2. The association between serum magnesium, phosphate and cognitive function may be closely related to the patients’ GI absorption and nutrition status. The authors should provide investigation of nutritional status by applying risk score such as Subjective global assessment (SGA) to distinguish whether the declined cognitive function was simply associated with poor nutrition status.
3. Certain symptomatic treatment agent which was commonly used in hemodialysis patients including pregabalin for uremic itch, muscle relaxant for muscle ache could largely affected patients’ cognitive function and could possibly be underestimated. The authors should provide data of possible offending drug use.
Comments on the Quality of English LanguageNA
Reviewer 3 Report
Comments and Suggestions for Authors
nutrients-3137438
Association between cognitive function and magnesium in patients undergoing hemodialysis: A cross-sectional analysis of the Osaka Dialysis Complication Study
This is an interesting cross-sectional clinical study, properly conducted, which aimed to analyze the associations of cognitive function and factors related to bone mineral disorder among 1207patients on maintenance hemodialysis.
However, the relevance of these results in the clinical practice of patients remains to be determined. Furthermore, some other comorbidities other than or additional to chronic kidney disease have not been taken into account.
The results shown in this study do not support a direct relationship between serum magnesium and serum phosphate levels with lower cognitive function, showing a clear absence of causality.
References corresponding to the year 2024 are missing.

Round 2
Reviewer 3 Report
Comments and Suggestions for Authors
nutrients-3137438
Association between cognitive function and magnesium in patients undergoing hemodialysis: A cross-sectional analysis of the Osaka Dialysis Complication Study
The authors have followed the suggestions.
However, the authors, following the suggestion of this reviewer, have consulted 2024 works, mainly the recently incorporated Reference 34 (Kato et al. Association between serum magnesium levels and cognitive function in patients undergoing hemodialysis. Clin Exp Nephrol. 2024.10.1007/s10157-024-02528-0) revealing the great loss of originality in this revised manuscript. Given the recent publications on this precise aspect, this could be a major drawback to the publication of this article.
